# Modelling SARS-CoV-2 Binding Antibody Waning 8 Months after BNT162b2 Vaccination

**DOI:** 10.3390/vaccines10020285

**Published:** 2022-02-13

**Authors:** Angelos Hatzakis, Andreas Karabinis, Sotirios Roussos, Nikos Pantazis, Dimitrios Degiannis, Antigoni Chaidaroglou, Konstantinos Petsios, Ioanna Pavlopoulou, Sotirios Tsiodras, Dimitrios Paraskevis, Vana Sypsa, Mina Psichogiou

**Affiliations:** 1Department of Hygiene, Epidemiology and Medical Statistics, Medical School, National and Kapodistrian University of Athens, 157 72 Athens, Greece; sotirisroussos@yahoo.co.uk (S.R.); npantaz@med.uoa.gr (N.P.); dparask@med.uoa.gr (D.P.); vsipsa@med.uoa.gr (V.S.); 2Hellenic Scientific Society for the Study of AIDS, Sexually Transmitted and Emerging Diseases, 115 27 Athens, Greece; 3Onassis Cardiac Surgery Center, 176 74 Athens, Greece; karabinis291@hotmail.com (A.K.); degiannis@yahoo.com (D.D.); achaidaroglou@yahoo.com (A.C.); petsiosk@gmail.com (K.P.); 4Pediatric Research Laboratory, Faculty of Nursing, National and Kapodistrian University of Athens, 157 72 Athens, Greece; idpavlop@yahoo.gr; 5Fourth Department of Internal Medicine, Attikon Hospital, Medical School, National and Kapodistrian University of Athens, 157 72 Athens, Greece; sotirios.tsiodras@gmail.com; 6First Department of Internal Medicine, Laiko General Hospital, Medical School, National and Kapodistrian University of Athens, 157 72 Athens, Greece; mpsichog@med.uoa.gr

**Keywords:** COVID-19, BNT162b2 vaccine, health care workers, immune response, anti-RBD, kinetics

## Abstract

Several lines of evidence suggest that binding SARS-CoV-2 antibodies such as anti-SARS-CoV-2 RBD IgG (anti-RBD) and neutralising antibodies (NA) are correlates of protection against SARS-CoV-2, and the correlation of anti-RBD and NA is very high. The effectiveness (VE) of BNT162b2 in preventing SARS-CoV-2 infection wanes over time, and this reduction is mainly associated with waning immunity, suggesting that the kinetics of antibodies reduction might be of interest to predict VE. In a study of 97 health care workers (HCWs) vaccinated with the BNT162b2 vaccine, we assessed the kinetics of anti-RBD 30–250 days after vaccination using 388 individually matched plasma samples. Anti-RBD levels declined by 85%, 92%, and 95% at the 4th, 6th, and 8th month from the peak, respectively. The kinetics were estimated using the trajectories of anti-RBD by various models. The restricted cubic splines model had a better fit to the observed data. The trajectories of anti-RBD declines were statistically significantly lower for risk factors of severe COVID-19 and the absence of vaccination side effects. Moreover, previous SARS-CoV-2 infection was associated with divergent trajectories consistent with a slower anti-RBD decline over time. These results suggest that anti-RBD may serve as a harbinger for vaccine effectiveness (VE), and it should be explored as a predictor of breakthrough infections and VE.

## 1. Introduction

Two years after the beginning of the COVID-19 pandemic and one year into the widespread application of COVID-19, there is extensive data on vaccine efficacy/effectiveness (VE) based on pivotal randomised trials and several ongoing vaccine effectiveness studies around the globe. VE assessment is based on several outcomes such as symptomatic disease, hospitalisation, severe disease, death, asymptomatic disease, or any documented infection [1].

The randomised trials of mRNA vaccines against COVID-19 illness initially provided data suggesting an efficacy of 95% for BNT162b2 and of 94.1% for mRNA-1273 at a median follow-up period of 2 months [2,3]. In the analysis of these trials 6–7 months after vaccination, the VE for BNT162b2 was 91.3% for COVID-19 illness and 96.7% against severe disease. For mRNA-1273, VE was 93.2% for COVID-19 illness, 98.2% for severe disease, and 63% for asymptomatic infections. Four months after the second dose, the VE was 83.7% for BNT162b2, while no change was noted for mRNA-1273 [4,5].

An analysis including Pfizer/BioNTech BNT162b2, Moderna mRNA-1273, AstraZeneca ChAdOx1-S, and Johnson & Johnson Ad26.COV2.S vaccines concluded that vaccination remains very effective in the prevention of severe disease/hospitalisation, with only an 8.0 percentage point (95% confidence interval (95% CI) 3.6–15.2) reduction in VE between 1 and 6 months from complete vaccination. However, the VE for symptomatic disease and infection declined by 25.4 (95% CI 13.7–42.2) and 18.5 percentage points (95% CI 8.4–33.4), respectively [1].

Among mRNA vaccines, mRNA-1273 was more effective than the BNT162b2 vaccine in all outcomes of SARS-CoV-2 [6]. This difference was more pronounced in preventing infection [7]. However, both vaccines were highly effective against severe/critical or fatal disease. The reduction in VE for infection was especially pronounced for BNT162b2, where it was found to be 20–47% after 5–7 months from the second dose [8,9]. The Delta variant had minimal effect on the VE against severe disease [10,11], and the waning VE was mainly attributed to waning immunity [1].

Several studies have documented a significant reduction in binding and neutralising antibodies up to 6–7 months from the second dose [12,13,14,15,16,17,18]. Antibody concentrations reach maximum levels in mRNA vaccines 2–3 weeks after the second dose and decline thereafter. Peak concentrations were found to be higher in females, in younger ages, in vaccinees without risk factors for COVID-19, in vaccinees experiencing adverse reactions after vaccination, and in vaccinees with previous SARS-CoV-2 infection [19,20,21,22,23].

VE and antibody titers wane over time, and it would be of interest to model the kinetics of antibody decline over time for use in studies aiming to predict VE and the risk of breakthrough infections. The pattern of antibody decline is not known, with a few studies suggesting a log-linear or power-law reduction [14,17,24].

In the current real-world study, we modelled the decline of anti-spike RBD (anti-RBD) SARS-CoV-2 8 months from vaccination or up to 7 months after the second dose of BNT162b2 using various models, and we assessed the waning trajectories according to demographic and clinical characteristics.

## 2. Material and Methods

### 2.1. Vaccination of Health Care Workers

Participants were vaccinated with 2 doses of BNT162b2 21 days apart. The study was designed to assess immunogenicity at time intervals of 1–2 weeks after the 2nd dose (28–35 days) and 4, 8, and 12 months after the 1st dose. Immunogenicity 1–2 weeks after the 2nd dose was expected to be highest, based on results from phase I/II studies [25]. In our previous study, we found that anti-RBD peaked 9–11 days after the 2nd dose [19].

Vaccinated health care workers (HCWs) from 2 teaching hospitals, Laiko General Hospital (Hospital 1) and Onassis Cardiac Surgery Center (Hospital 2), participated in the immunogenicity study after signing informed consent (*n* = 871) [19]. Participants in this study comprised a group of HCWs from Onassis Cardiac Surgery Center who consented to donate an additional sample at the 6th month. All HCWs in this group (*n* = 97) provided samples at months 1, 4, 6, and 8 since their 1st vaccine dose.

A brief questionnaire was administered to HCWs concerning information about age, gender, education, position within the hospital, body mass index (BMI), history of risk factors for severe COVID-19 (RFS-CoV), previous COVID-19 (Pr-CoV), and history of self-reported adverse reactions after vaccination (VSEs). VSEs were grouped according to the major symptoms such as local pain, fever, fatigue, and allergic and other systematic reactions. Combinations of VSEs were counted as one VSE in the final analysis [19]. RFS-CoV included hypertension, diabetes mellitus, cardiovascular disease, obesity, malignant neoplasm, chronic renal disease, chronic liver disease, and immunosuppressive treatment [19].

### 2.2. Serological Tests

Serum samples collected after venipuncture were tested for SARS-CoV-2 IgG binding antibodies to nucleocapsid protein (anti-N) and SARS-CoV-2 receptor-binding domain spike protein IgG. Both assays were described in detail in our previous publication [19].

The first assay was a qualitative one, with an index (sample/calibrator (s/c)) cutoff of 1.4. Samples with an index ≥1.4 are considered positive and <1.4 negative. Reactive results are considered as an indication of natural infection.

The second assay (Abbott SARS-CoV-2 IgG II Quant) or anti-RBD was used to quantify IgG antibodies against the receptor-binding domain (RBD) of the S1 subunit of the spike protein. The linear range is between 21 and 40,000 AU/mL, with a cutoff value of 50 AU/mL. Both assays are based on chemiluminescent microparticle immunoassay (CLIA) [26].

The correlation coefficient of Abbott anti-RBD with the World Health Organization (WHO) standard is 0.999, and the transformation of Abbott anti-RBD AU/mL to WHO BAU/mL is possible by using the equation BAU/mL = 0.142 × AU/mL [26].

### 2.3. Statistical Analysis

Continuous variables were described using mean and standard deviation, or median, 25th, and 75th percentiles. Antibody titers were presented as geometric mean titers (GMTs) with the corresponding 95% CI. Categorical variables were described using frequencies and percentages. Chi-squared and Mann–Whitney U tests were used to assess differences in demographics and clinical characteristics among the participants of this study (*n* = 97) and the remaining ones (*n* = 774).

Exploratory data analysis for anti-SARS-CoV-2 RBD antibodies was based on available measurements within 26–258 days after the first dose of the vaccine or 5–237 days after the second dose. We used log_10_ transformation in anti-SARS-CoV-2 RBD antibodies to normalise the distribution.

Longitudinal changes in antibody levels were analysed through three linear mixed models: (a) exponential model (EM), (b) power-law model (PLM), and (c) mixed model using four-knot restricted cubic splines (restricted cubic splines model) (RCSM), assuming that the peak of antibody titers was at Study Day 30 (i.e., 9 days after the second dose) [19]. Linear mixed models have been well studied within the classical/likelihood [27] and Bayesian [28]. frameworks. Typical applications of EM, PLM, and RCSM mixed models can be found in [14]. and [29]. Half-lives (i.e., time after peak required to reach antibody titers equal to ½ of the peak levels) were calculated based on the three previous models [14,27,28,29].

#### 2.3.1. Exponential Model (EM)

The general form of the exponential model takes the following form:(1)$$\log_{10}\left(Titer_{i,j}\right)=\left(\beta_{0}+b_{0i}\right)+\left(\beta_{1}+b_{1i}\right)\cdot \left(study\;day_{i,j}\right)+e_{i,j}$$
where $\beta_{0}$ and $\beta_{1}$ are the fixed effects, intercept, and decay rate (slope), respectively. $b_{0i}$ and $b_{1i}$ are the random effects, intercept, and decay rate for each participant, and $e_{i,j}$ is the model errors for participant i at study day j. Hence, log_10_ transformation of titers is a linear function of time (i.e., steady decay rate in log scale over time) [14,30,31]. The half-life $\left(t_{1/2}\right)$ is given by the following equation:(2)$$t_{1/2}=-\frac{\log_{10}\left(2\right)}{\beta_{1}}$$
and the 95% confidence interval of half-life was calculated using the delta method.

#### 2.3.2. Power-Law Model (PLM)

The general form of the power-law model takes the following form:(3)$$\log_{10}\left(Titer_{i,j}\right)=\left(\beta_{0}+b_{0i}\right)+\left(\beta_{1}+b_{1i}\right)\cdot \log_{10}\left(\left(study\;day_{i,j}\right)-21\right)+e_{i,j}$$

Where $\beta_{0}$ and $\beta_{1}$ are the fixed effects, intercept, and decay rate (slope), respectively. $b_{0i}$ and $b_{1i}$ are the random effects, intercept, and decay rate for each participant, and $e_{i,j}$ is the model errors for participant i at study day j. Study day was offset by 21 days to account for the 2nd dose regimen. Hence, the log_10_ transformation of titers is a linear function of the log_10_ transformation of time (i.e., decay rates decrease over time) [14,31]. The half-life is given from the following form:(4)$$t_{1/2}=10^{\log_{10}\left(9\right)-\frac{\log_{10}\left(2\right)}{\beta_{1}}}-9$$
and the 95% confidence interval of half-life was calculated using the delta method.

#### 2.3.3. Restricted Cubic Splines Model (RCSM)

A restricted cubic spline is a set of piecewise cubic functions (polynomials), where the boundaries of these pieces are called knots. The curves pass through all the knots, and both first- and second-order derivatives (the slope and the rate of slope, respectively) are the same for both functions on either side of a knot. Restricted splines are constrained to be linear beyond boundary knots (i.e., before the first knot and after the last one) [32].

Restricted cubic splines with four knots were used to estimate the time kinetics curves of antibodies. The location of the knots was placed in the percentiles recommended by Harrell [33]. The general form of the mixed model with a four-knot restricted cubic spline takes the following form [32]:(5)$$\log_{10}\left(Titer_{i,j}\right)=\beta_{0}+\sum_{k=1}^{3}\beta_{k}\cdot S_{k}\left(t\right)+b_{0i}+b_{1i}\cdot s_{1}\left(t\right)+b_{2i}\cdot s_{2}\left(t\right)+e_{i,j}$$
where $b_{0i}$ is the random intercept for each participant, $e_{i,j}$ is the model errors for participant i at study day j, $S_{k}\left(t\right)$, k=1,2,3 are restricted cubic splines with four-knot terms, $s_{1}\left(t\right)$, $s_{2}\left(t\right)$ are restricted cubic splines with three-knot terms, and $b_{1i}$, $b_{2i}$ are the corresponding random effects. The half-life and the corresponding 95% confidence interval of were estimated using a Monte Carlo procedure.

#### 2.3.4. Model Selection

Log-likelihood is a measure of model fit. Higher values indicate better fit. To compare the fit of the evaluated models, we used the likelihood ratio test and the AIC (Akaike information criterion), which takes the following form:(6)AIC=−2⋅(log−likelihood)+2⋅k
where k is the number of model parameters. Lower values of AIC indicate a better fit.

The best-fitted model was applied to assess relative differences in means of anti-RBD IgG levels by gender, age, risk factors for severe COVID-19 illness, side effects of vaccination, and history of previous COVID-19 infection in the post-vaccination period (5–237 days after the 2nd dose).

All analyses were performed using Stata version 13.0 [33]. All *p*-values were two-sided, and *p* < 0.05 was considered statistically significant.

## 3. Results

Ninety-seven vaccinated HCWs, donating 388 blood samples, participated in all rounds of blood sampling. Their demographic and clinical characteristics are shown in Table 1. The majority were females (55.7%), Greek nationals (93.8%), highly educated holding an MSc, Ph.D., or MD degree (40.2%), treating or caring patients (67%) with the mean (SD) age of 50.1 (9.4) years and BMI of 26.4 (5.5) kg/m^2^, including 21.6% obese, 15.5% reporting RFS-CoV, 54.6% VSEs, and 7.2% with Pr-CoV.

Overall, the distribution of demographic and clinical characteristics of 97 HCWs was similar to the distribution of 774 of the remaining HCWs, who participated in our previous study [19], with the exception of HCWs involved in patient care who were more frequent in the studied population (83% vs. 67%) (Table 1).

In Appendix A, the timing of blood samplings is shown. The median (25th–75th) times were 30 (29–32), 125 (124–126), 183 (182–185), and 251 (250–252) days from the first vaccine dose to first, second, third, and fourth measurement, respectively. The corresponding times from the second vaccine dose were 9 (8–10), 103 (102–105), 161 (160–163), and 229 (228–231) days, respectively.

Overall, the GMTs (95% CI) at times 30, 125, 183 and 251 were 13,674 (11,164–16,749) AU/mL, 2112 (1716–2601) AU/mL, 1036 (834–1288) AU/mL, and 617 (490–778) AU/mL, respectively. The percentage reduction in GMTs of anti-RBD levels from the 30th day to the 125th, 183rd, 251st days was 85%, 92% and 95%, respectively. The percentage reductions were similar when we used the median of anti-RBD levels (86%, 93%, and 96%, respectively) (Table 2, Appendix A).

We further explored this significant waning of anti-RBD by studying the pattern of waning. We used three models of reduction: EM, PLM, and RCSM. Based on the AIC criterion, the best fitting was the RCSM with AIC values −202.4 compared with 155.8 and 58.5 for EM and PLM, respectively (Table 3, Figure 1).

According to RCSM, the day where antibody levels reached 50% reduction from the peak levels of the 30th day was day 62 (95% CI 59.9–64.5), while in EM and PLM, this was the 79th (95% CI 76–81) day and the 40th (95% CI 39–41) day, respectively (Table 3).

The GMTs of anti-RBD levels trajectories are presented by gender, age, risk factors for severe COVID-19 illness, side effects of vaccination, and history of COVID-19 infection (Table 4).

By using the RCSM mixed-effects model, we estimated differences in anti-RBD in the trajectory from 30 up to 251 days from the first dose. In the univariable and multivariable analysis, age and sex were not associated with difference in the decline trajectory. The presence of RFS-CoV was significantly associated with decreasing anti-RBD levels by 43.6% (95% CI 5.1–66.7%, *p* = 0.031). A lack of VSEs was associated with decreasing levels of anti-RBD throughout the immunogenicity trajectory by 38.6% (95% CI 11.9–57.2%, *p* = 0.001). The aforementioned effects did not show any statistically significant variation over time (i.e., interactions with time terms were not significant), with the exception of previous SARS-CoV-2 infection, the effect of which on anti-RBD levels increased with time (interaction with time terms *p*-value = 0.019). The increase ranged from 20.3% to 205.8%, resulting in lesser waning in those vaccinated with Pr-CoV (Table 4, Figure 2).

## 4. Discussion

There is overwhelming evidence that neutralising (NA) and binding antibodies such as anti-spike SARS-CoV-2 IgG (anti-spike) and anti-RBD SARS-CoV-2 IgG are immune correlates of protection (Cor-P). Cor-P are immunological markers that can be used to reliably predict the level of vaccine efficacy in a clinically relevant protection endpoint or the level of protection against breakthrough infections in vaccinated individuals [24,34,35,36,37,38,39,40,41,42]. Cor-P measured longitudinally in a vaccinated cohort may inform on the waning of vaccine efficacy, the timing of potential anamnestic dose, or the timing to update vaccine composition due to the emergence of new variants. Moreover, this may allow new vaccines to be authorised for use on the basis of immunogenicity and safety alone when large efficacy trials are not feasible. Cor-P have been established against many viral diseases [43].

Immune correlates of protection are meaningful for protection against symptomatic COVID-19 but not against asymptomatic infection [37,42]. Studies have shown increasing levels of VE associated with increasing immune marker levels. However, a threshold for protection was not found [37,42]. Among the immune markers examined, anti-RBD and anti-spike have a similar correlation with NA in predicting VE [37,42]. This finding has major implications for monitoring the VE, since NA testing is complex and not widely available. Sparse data from the literature suggest a reduction in VE for BNT162b2 against asymptomatic infection from 73% 1 month after the 2nd dose to 48% and 24% at months 3 and 5, respectively [8]. This pattern of VE reduction against infection follows the anti-RBD reduction pattern observed in this and other studies [1,12,13,17,44]. Therefore, the monitoring of binding antibodies in a vaccinated cohort may serve as a harbinger of waning VE, and it may highlight the need for a booster dose.

The three models applied to assess waning humoral immunity belong to the same family of mixed linear models. The only difference between the three models is related to the treatment of time as entered in the fixed and random effects (i.e., untransformed in the EM, log transformed in the PLM, and through spline functions in the RCSM). Existence, estimation, and properties of the solutions of mixed linear models are well studied within the classical/likelihood based framework and the Bayesian frameworks [27,28]. Such models have been applied numerous times in biomedical research when the evolution of a continuous marker over time is of interest and repeated measurements per individual are available. Typical examples include the evolution of CD4 cell count or HIV-RNA viral load levels in HIV-infected individuals, prostate-specific antigen (PSA) levels in prostate cancer patients, BMI evolution in children, respiratory parameters in chronic obstructive pulmonary disease (COPD) patients, etc. A typical example of a mixed linear model with restricted cubic splines can be found in the work of Mallon PW et al. [29].

Previous studies on antibody waning modelling with Moderna mRNA-1273 and Pfizer BNT162b2 vaccines used EM or PLM, respectively [14,17,24]. Our current analysis suggests that these models cannot accurately predict the antibody waning pattern, and an RCSM is more appropriate. A mixed-effects model with a restricted cubic spline has the flexibility to capture the more complex relationship between the variables, as well as a rich structure in random effects using a restricted cubic spline with four knots. Thus, better modelling was achieved both for the mean of anti-RBD levels and their variability. However, a large study including more knots may provide more accurate prediction of waning pattern.

The study of antibody trajectories is more appealing than using individual time points to study the demographic and clinical characteristics that might be associated with antibody levels. Based on RCSM, we examined the post-vaccination anti-RBD trajectories 3–251 days after the first dose according to gender, age, RFSC, VSE, and Pr-CoV. The gender and age trajectory differences did not differ in a statistically significant way. Statistically significant differences in the trajectories of anti-RBD were found for RFS-CoV, VSE, and Pr-CoV. Previous studies documented that after completion of the second dose of BNT162b2, those not reporting RFS-CoV and VSE had higher anti-RBD levels [12,23,44]. A finding of our study is the increasing anti-RBD difference over time according to previous COVID-19 infection, which suggests a faster decline in vaccinated individuals with no prior infection compared with Pr-CoV [45]. The waning of antibodies in Pr-CoV is consistent with a large study from Qatar, where the cumulative incidence of breakthrough infections was 5.5 times higher in vaccinated individuals with no prior COVID-19 infection compared with Pr-CoV ones [46]. Several studies suggest that the quality of immune response in vaccinated Pr-CoV improves over time in terms of neutralisation capacity and breadth (hybrid immunity), with the improved neutralisation of new variants including VOC Omicron, especially after the use of booster dose [47,48,49,50,51,52,53,54,55,56,57,58].

The limitations of our study include a restricted age range and a small sample size for a thorough assessment of anti-RBD trajectories. Previous studies have shown a sharp reduction in anti-RBD after the age of 65 years [12,54]. Additionally, humoral immunity studies may not adequately capture SARS-CoV-2 immunity due to the lack of data on cellular immunity such as B- and T-cell SARS-CoV-2-specific markers.

## 5. Conclusions

Overall, we conclude that (1) the profound decline of anti-RBD 4–8 months after the first dose is a harbinger of a reduction in vaccine effectiveness to prevent SARS-CoV-2 infection, (2) the use of a more precise PCSM for an assessment of the waning pattern will facilitate modelling studies to evaluate the determinants of humoral immunity patterns, and (3) anti-RBD should be explored as a predictor for breakthrough infections and as a strategy to simplify the assessment of vaccine efficacy for new vaccines [36,59].

## Figures and Tables

**Figure 1 vaccines-10-00285-f001:**
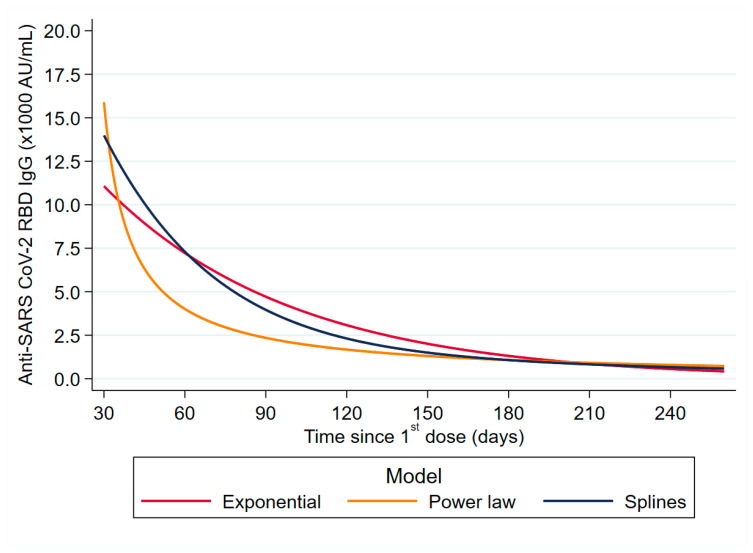
Comparison of three models (splines, exponential, and power-law) for the predicted trajectories of anti-SARS-CoV-2 RBD IgG antibody levels over time.

**Figure 2 vaccines-10-00285-f002:**
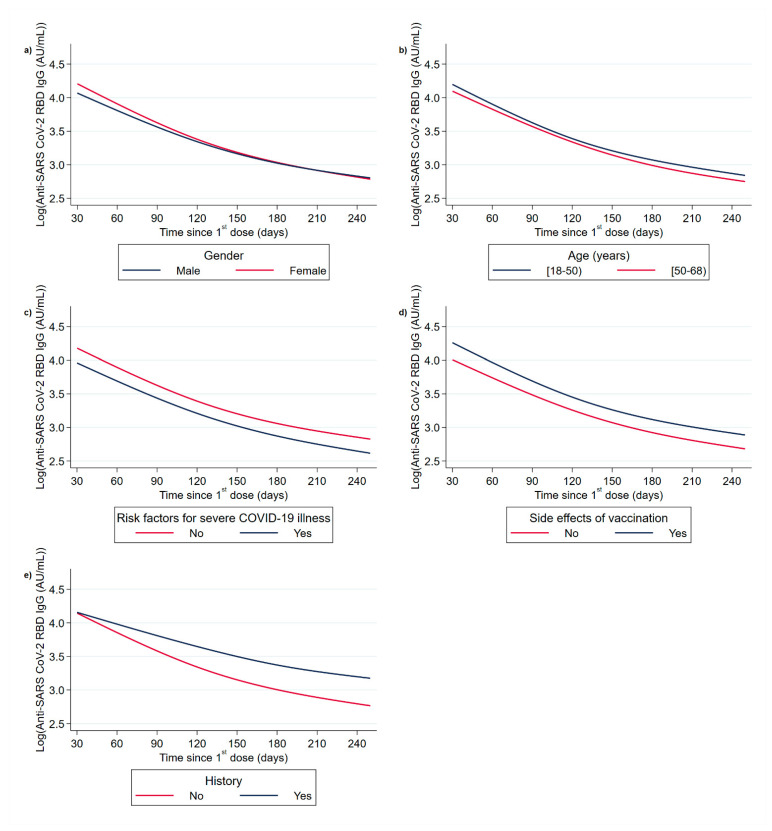
Waning, based on mixed model, of anti-SARS-CoV-2 RBD IgG antibody levels over time from the first dose by (**a**) gender, (**b**) age, (**c**) risk factors for severe COVID-19 illness, (**d**) side effects of vaccination, and (**e**) history of COVID-19 infection in pre-vaccination period.

**Table 1 vaccines-10-00285-t001:** Demographic and clinical characteristics of study participants.

Covariate	Total *n* = 97	Total *n* = 774	*p*-Value
Gender, n (%)			0.090 ^1^
Male	43 (44.3)	275 (35.5)	
Female	54 (55.7)	499 (64.5)	
Age (years), mean (SD)	50.3 (9.4)	47.4 (10.4)	0.011 ^2^
Age (years), median (25th–75th)	50.1 (46.1–56.6)	48.9 (39.2–55.2)	0.019 ^3^
Age (years), n (%)			0.349 ^1^
[18–50)	47 (48.4)	414 (53.5)	
[50–68)	50 (51.6)	360 (46.5)	
Country of birth, n (%)			0.555 ^1^
Greece	91 (93.8)	713 (92.1)	
Other	6 (6.2)	61 (7.9)	
Body mass index (BMI) (kg/m^2^), mean (SD)	26.4 (5.5)	26.2 (4.8)	0.653 ^2^
Body mass index (BMI) (kg/m^2^), median (25th–75th)	25.6 (22.9–29.1)	25.7 (22.8–29.0)	0.988 ^3^
Body mass index (BMI) (kg/m^2^), n (%)			0.770 ^1^
Under/Normal weight: <25	41 (42.3)	342 (44.2)	
Overweight: 25–30	35 (36.1)	288 (37.2)	
Obesity: ≥30	21 (21.6)	144 (18.6)	
Education, n (%)			0.136 ^1^
High school or below	28 (28.9)	179 (23.2)	
University	30 (30.9)	319 (41.3)	
MSc-Ph.D.	39 (40.2)	275 (35.6)	
Health care workers, n (%)			<0.001 ^1^
Yes	65 (67.0)	644 (83.2)	
No	32 (33.0)	130 (16.8)	
Risk factors for severe COVID-19 illness, n (%)			0.960 ^1^
Yes	15 (15.5)	119 (15.7)	
No	82 (84.5)	641 (84.3)	
Side effects of vaccination, n (%)			0.626 ^1^
Yes	53 (54.6)	443 (57.2)	
No	44 (45.4)	331 (42.8)	

^1^ Chi-squared test; ^2^ Student’s *t*-test; ^3^ Mann–Whitney U test.

**Table 2 vaccines-10-00285-t002:** Geometric mean with 95% confidence interval (AU/mL) concentration of anti-SARS-CoV-2 RBD IgG antibodies at four time points after the first dose of BNT162b2 vaccine by different characteristics, ***n*** = 97.

Covariate		Study Day 30		Study Day 125		Study Day 183		Study Day 251	
	N	Geometric Mean (95% CI)	*p* ^1^	Geometric Mean (95% CI)	*p* ^1^	Geometric Mean (95% CI)	*p* ^1^	Geometric Mean (95% CI)	*p* ^1^
Overall	97	13,674 (11,164–16,749)		2112 (1716–2601)		1036 (834–1288)		617 (490–778)	
Gender			0.135		0.743		0.918		0.856
Male	43	11,520 (7974–16,641)		2032 (1400–2950)		1023 (696–1504)		632 (424–942)	
Female	54	15,674 (12,558–19,564)		2179 (1716–2767)		1047 (812–1350)		606 (458–801)	
Age (y)			0.258		0.577		0.355		0.359
[18─50)	47	15,413 (10,917–21,763)		2245 (1609–3131)		1151 (817–1622)		690 (483–984)	
[50─68)	50	12,218 (9696–15,396)		1995 (1530–2602)		939 (708–1245)		556 (409–757)	
Risk factors for severe COVID-19 illness			0.072		0.181		0.158		0.133
Yes	15	8894 (3493–22,647)		1520 (588–3928)		721 (278–1871)		410 (154–1092)	
No	82	14,793 (12,362–17,702)		2243 (1859–2707)		1107 (905–1355)		666 (536–827)	
Adverse effects of vaccination			0.003		0.042		0.043		0.042
Yes	53	17,983 (14,111–22,918)		2563 (1950–3368)		1268 (952–1689)		765 (568–1032)	
No	44	9831 (7125–13,565)		1674 (1219–2299)		813 (584–1131)		477 (333–682)	
History of COVID-19 infection up to the 2nd dose			0.912		0.071		0.044		0.036
Yes	7	14,246 (2396–84,696)		4166 (525–33,070)		2283 (254–20,538)		1478 (154–14,173)	
No	90	13,631 (11,247–16,519)		2004 (1667–2409)		974 (806–1179)		577 (470–708)	

^1^ Student’s *t*-test; abbreviation: CI, confidence interval.

**Table 3 vaccines-10-00285-t003:** Days required to halve anti-SARS-CoV-2 RBD IgG antibody levels after the first dose of the BNT162b2 compared to peak levels (i.e., at 30 days). Estimates from different models.

Model	Ν	Log-Likelihood	df	AIC	Study Day of t_1/2_	95% CI
Exponential	97	−71.9	6	155.8	78.7	76.1–81.4
Power-Law	97	−23.3	6	58.5	39.8	39.0–40.6
Restricted cubic spline	97	112.2	11	−202.4	62.0	59.9–64.5

Abbreviation: df, degrees of freedom; AIC, Akaike information criterion; CI, confidence interval.

**Table 4 vaccines-10-00285-t004:** Relative differences in means of anti-SARS CoV-2 RBD IgG levels by gender, age, risk factors for severe COVID-19 illness, side effects of vaccination, and history of previous COVID-19 infection in the BNT162b2 post-vaccination period. Results based on univariable and multivariable analysis based on mixed models including restricted cubic spline time terms.

		Univariable			Multivariable		
Covariate	N	% Difference	95% CI	*p*-Value	% Difference	95% CI	*p*-Value
Gender ^1^							
Male	43	Ref.					
Female	54	26.1%	−13.7–84.4%	0.230			
Age (y) ^2^							
[18–50)	47	Ref.					
[50–68)	50	−16.2%	−42.6–22.3%	0.360			
Risk factors for severe COVID-19 illness ^3^							
No	82	Ref.			Ref.		
Yes	15	−36.6%	−62.2–6.3%	0.084	−43.6%	−66.6–−5.1%	0.031
Side effects of vaccination ^4^							
Yes	53	Ref.			Ref.		
No	44	−40.4%	−58.8–−14.0%	0.006	−38.6%	−57.2–−11.9%	0.001
History of COVID-19 infection ^5^							
No	90	Ref.			Ref.		
Yes (at 29 days)	7	1.6%	−52.9–119.2%	0.968	20.3%	−44.0–158.4%	0.635
Yes (at 125 days)		106.2%	-4.1–343.3%	0.064	144.4%	13.0–428.3%	0.023
Yes (at 183 days)		134.0%	3.4–429.7%	0.041	177.2%	21.8–530.5%	0.015
Yes (at 253 days)		158.0%	9.5–508.1%	0.030	205.8%	30.1–618.6%	0.010

^1^ Interaction with time terms non-significant (likelihood ratio (LR) test *p* = 0.141); ^2^ interaction with time terms non-significant (LR test *p* = 0.096); ^3^ interaction with time terms non-significant (likelihood ratio (LR) test *p* = 0.783); ^4^ interaction with time terms non-significant (likelihood ratio (LR) test *p* = 0.518); ^5^ interaction with time terms significant (likelihood ratio (LR) test *p* = 0.019), estimated relative differences given at specific time points (i.e., 29, 125, 183, and 253 days after the 1st vaccine dose); abbreviation: CI, confidence interval.

## Data Availability

All relevant data are available at the Pergamos Institutional Repository of the National and Kapodistrian University of Athens, Greece. A link will be available upon review of the manuscript.

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
