# Peer review of "Modelling SARS-CoV-2 Binding Antibody Waning 8 Months after BNT162b2 Vaccination"

_vaccines, 2022, doi:10.3390/vaccines10020285_

Round 1

Reviewer 1 Report

Hatzakis et al. assessed the kinetics of anti-RBD after vaccination and established the model fits kinetics. The manuscript is clear and reasonably insightful. There are a few instances where they overstep their data and a few minor concerns, but I find no major flaws in their methodologies or interpretation. In general, the studies are well-performed and an advance for the COVID19 research field. I have only a few suggestions for the authors to consider.

  1. Line 96, did the anti-N decline trajectory also fit the model established for anti-RBD? Meanwhile, are there any advances for monitoring anti-RBD instead of anti-spike whole protein. There are reports suggested that antibody targeting N terminal of S protein could also have neutralizing capability.
  2. Line 97, receptor-binding domain (RBD) should be moved to line 69 where it appears first in introduction section.
  3. The authors observed that lack of VSEs was associated with decreasing levels of anti-RBD, did participants with VSE had higher initial anti-RBD levels?
  4. Line 263, are there other method suitable for monitoring anti-RBD or anti-spike levels? How prevalent CLIA is used in clinics or institutions?
  5. Line 29, “These results suggest that anti-RBD may serve as a harbinger for vaccine effectiveness (VE)” Although authors reported that the presence of RFS-CoV was significantly associated with decreasing anti-RBD level, no neutralization test was performed in this study. I would suggest authors modify the sentence in line 29, which is stated well in 266-268.

Author Response

Responses to Reviewers

Reviewer 1.

Comment 1: Line 96, did the anti-N decline trajectory also fit the model established for anti-RBD? Meanwhile, are there any advances for monitoring anti-RBD instead of anti-spike whole protein. There are reports suggested that antibody targeting N terminal of S protein could also have neutralizing capability.

Response: In the Discussion is presented an extensive list of references (32-40) supporting that antibodies against receptor binding domain (RBD, against the full spike and NA are excellent correlates of protection. The correlation between anti-spike and anti-RBD exceeds 0.95. Therefore, anti-RBD is widely used in immunogenicity studies.

---------------------------------------------------------------------------------------------------------------------------

Comment 2: Line 97, receptor-binding domain (RBD) should be moved to line 69 where it appears first in introduction section.

Response: We corrected according to the suggestion.

---------------------------------------------------------------------------------------------------------------------------

Comment 3: Line 97, receptor-binding domain (RBD) should be moved to line 69 where it appears first in introduction section.

Response: As shown in Fig 2d and mentioned in the text (“Lack of VSEs was associated with decreasing levels of anti-RBD throughout the immunogenicity trajectory by 38.6% (95% CI 11.9%-236 57.2%, p=0.001). The aforementioned effects did not show any statistically significant variation over time …”). Indeed participants with VSEs had higher initial/peak (i.e. at 30 days after 1st dose) anti-RBD levels and retained this difference throughout the whole follow-up time.

--------------------------------------------------------------------------------------------------------------------------

Comment 4: Line 263, are there other method suitable for monitoring anti-RBD or anti-spike levels? How prevalent CLIA is used in clinics or institutions?

Response: The method used for anti-RBD testing is chemiluminescent microparticle immunoassay (CLIA). The testing is based on an automated platform provided by Abbott Laboratories and is available globally.

---------------------------------------------------------------------------------------------------------------------------

Comment 5: Line 29, “These results suggest that anti-RBD may serve as a harbinger for vaccine effectiveness (VE)” Although authors reported that the presence of RFS-CoV was significantly associated with decreasing anti-RBD level, no neutralization test was performed in this study. I would suggest authors modify the sentence in line 29, which is stated well in 266-268.

Response: The abstract was extensively modified to address this comment.

Reviewer 2 Report

Comments to the Authors:

  1. Abstract is incomplete: What motivation authors get to do such study? Objective is not clearly defined.
  2. The discussion section (Introduction) in the present form is relatively weak and should be strengthened with more details and justifications.For more update about the topic kindly refer the below articles :
  •  Atangana–Baleanu derivative with fractional order applied to the gas dynamics equations
  1. Authors have used Longitudinal changes in antibody levels were analyzed through three mixed models: Exponential model (EM), b) power-law model (PLM) and c) mixed model using four-knot restricted cubic splines (restricted cubic splines model) (RCSM). Please provide the existence of the solution of these three models and also provides some real world application.
  2. Please polish the grammar.
  3. Please provide the conclusion section of the article.

Author Response

Responses to Reviewers

Reviewer 2

Comment 1: Abstract is incomplete: What motivation authors get to do such study? Objective is not clearly defined.

Response: The abstract was substantially modified according to the Reviewer’s suggestion (see Abstract).

---------------------------------------------------------------------------------------------------------------------------

Comment 2+3: The discussion section (Introduction) in the present form is relatively weak and should be strengthened with more details and justifications. For more update about the topic kindly refer the below articles:

  • Atangana–Baleanu derivative with fractional order applied to the gas dynamics equations

Authors have used Longitudinal changes in antibody levels were analyzed through three mixed models: Exponential model (EM), b) power-law model (PLM) and c) mixed model using four-knot restricted cubic splines (restricted cubic splines model) (RCSM). Please provide the existence of the solution of these three models and also provides some real world application.

Response: We added paragraphs in the Methods and Discussion on the use of EM, PLM and RCSM and we provided some real world studies (lines 142-145 and 268-281).

The Atangana–Baleanu is used in a different context and probably is not useful for this paper.

---------------------------------------------------------------------------------------------------------------------------

Comment 4: Please polish the grammar.

Response: Grammar was corrected.

---------------------------------------------------------------------------------------------------------------------------

Comment 5: Please provide the conclusion section of the article.

Response: The conclusion was included at the end of the Discussion.

---------------------------------------------------------------------------------------------------------------------------

Reviewer 3 Report

The study is well structured and the selection of samples is representative, although as reported, by the authors, the sample size is not high and was considered a restricted age range.The analysis using “Restricted cubic splines” is certainly a flexible tool to analyze this complex model but  the conclusions must be reformulated.

It is a simplistic assessment that requires a model definitely more complex and not just three knots.

It’s known that the immune system is not limited to the mere presence and persistence of antibodies in the bloodstream, but also to the ability and responsiveness of their synthesis. Although the amount of antibodies in the blood decreases over the months, Immunological memory of B cells (specific for the Spike protein of SARS-Cov-2) and T lymphocytes lasts longer in time (presumably even longer than 8 months).

Author Response

Responses to Reviewers

Reviewer 3

Comment 1: It is a simplistic assessment that requires a model definitely more complex and not just three knots.

Response: The splines model in its basic form (i.e. with any covariates effects) is a 11 parameter model compared to the 6 parameters used in the Exponential and Power Law models. Thus, for the evolution of the average anti-RBD levels a four knots spline is used (i.e. 3 time terms) (compared to one term in the Exponential and Power Law models).In addition, the variance-covariance structure a 3x3 covariance matrix is used compared to the 2x2 matrices used in the other two models. Given the above we believe that the model has a very high degree of flexibility, it is definitely much more flexible than the typically used exponential or power law models and this flexibility is clearly supported by the likelihood ratio tests between the three models. Additionally, more flexible models have been assessed but their use, was not supported by the relevant tests. However, as a general precaution, we included in lines 289-290 the phrase “ However, a large study including more knots may provide more accurate prediction of waning pattern”.

---------------------------------------------------------------------------------------------------------------------------Comment 2: It’s known that the immune system is not limited to the mere presence and persistence of antibodies in the bloodstream, but also to the ability and responsiveness of their synthesis. Although the amount of antibodies in the blood decreases over the months, Immunological memory of B cells (specific for the Spike protein of SARS-Cov-2) and T lymphocytes lasts longer in time (presumably even longer than 8 months).

Response: The Reviews is right. We included his comment as a limitation of our study (lines 312-314).

Round 2

Reviewer 3 Report

The changes made have improved the comprehension of the text.

Remove by the conclusions "vaccine failure" because the decrease of RDB antibodies is not the only parameter of vaccine effectiveness. The cellular response is not taken into account as reported in sentence 307-309.

Remove round brackets from citations in the sentence line 142.

Author Response

Thank you for your suggestions which have been addressed. Very few minor spellings were corrected.

With kind regards,

Angelos Hatzakis